# DexMachina: Functional Retargeting for Bimanual Dexterous Manipulation

Zhao Mandi [1]  Yifan Hou [1]  Dieter Fox [2]  Yashraj Narang [2]  Ajay Mandlekar* [2]  Shuran Song* [1]

## Abstract

We study the problem of functional retargeting: learning dexterous manipulation policies to track object states from human hand-object demonstrations. We focus on long-horizon, bimanual tasks with articulated objects, which are challenging due to large action space, spatiotemporal discontinuities, and the embodiment gap between human and robot hands. We propose DexMachina, a novel curriculum-based algorithm: the key idea is to use virtual object controllers with decaying strength: an object is first driven automatically towards its target states, such that the policy can gradually learn to take over under motion and contact guidance. We release a simulation benchmark with a diverse set of tasks and dexterous hands, and show that DexMachina significantly outperforms baseline methods. Our algorithm and benchmark enable a functional comparison for hardware designs, and we present key findings informed by quantitative and qualitative results. With the recent surge in dexterous hand development, we hope this work will provide a useful platform for identifying desirable hardware capabilities and lower the barrier for contributing to future research. Videos and more at: `project-dexmachina.github.io`

## 1 Introduction

Dexterous robot hands, with their resemblance to human hands, spark the expectation for achieving human-level dexterity. Yet the reality presents many hardware and algorithmic challenges that bottleneck the progress in dexterous manipulation. Prior learning-based methods have seen success in relatively simple and short-horizon tasks, but are often limited by manual reward-engineering (Andrychowicz et al., 2020; Lum et al., 2024) or costly data-collection (Qin

et al., 2023; Andrychowicz et al., 2020) due to the embodiment gap between human hands and dexterous hands.

Human hands are hence a natural source for learning guidance. In this work, we formulate learning from human with an emphasis on task capability. We denote the problem as *functional retargeting*: given a human demonstration, the goal is to learn dexterous hand policies that can manipulate the object to follow the demonstrated trajectory (see Fig. 1). This is distinguished from *kinematic* retargeting (Qin et al., 2023), which produces human-like motions without ensuring feasibility. The problem is even more compelling for long-horizon, bimanual demonstrations with articulated objects, which encompass a significant portion of daily human activities, but pose several key challenges: exploration is difficult under the high-dimensional action space, the intricate contact sequences demand stable and precise hand movements; due to the embodiment gap, human hand motion cannot be directly mapped to feasible robot actions, which limits the scalability of imitation data collection.

To address these challenges, we propose DexMachina[1], a novel curriculum-based RL algorithm for functional retargeting. Precise bimanual coordination is often required to manipulate an object successfully (e.g. opening a waffle iron mid-air, see Fig. 1), but naive approaches often get stuck in early failures or suboptimal actions. This motivates us to design a curriculum to allow the policy to explore in a less fragile setting. Our key idea is to use *virtual object controllers*—they apply control forces that drive an object towards its demonstrated trajectory—and *auxiliary motion and contact rewards*, which guide the policy to learn task strategies as the virtual controller strength decays. The policy first learns to mimic the human motion without worrying about failing the task, then learns to take over manipulation as the virtual controllers fade away.

Despite continuous effort in developing new hands and sensing capabilities (Rakić, 1968; Loucks et al., 1987; Jacobsen et al., 1986; Butterfass et al., 1998; Shiokata et al., 2005; Higo et al., 2018), there is a lack for standardized and accessible evaluation benchmarks. To address this, we build a simulation benchmark with a diverse set of 6 dexterous

* Equal Advising [1]Department of Electrical Engineering, Stanford University, Palo Alto, U.S.A [2]NVIDIA, Santa Clara, U.S.A. Correspondence to: Zhao Mandi <mandi@stanford.edu>.

*Proceedings of the $43^{rd}$ International Conference on Machine Learning*, Seoul, South Korea. PMLR 306, 2026. Copyright 2026 by the author(s).

[1]Deus ex machina ("god from the machine"), is when a seemingly unsolvable problem is conveniently solved by an external force — much like how our algorithm moves an object by itself before the policy gradually learns to take over, hence the name DexMachina.

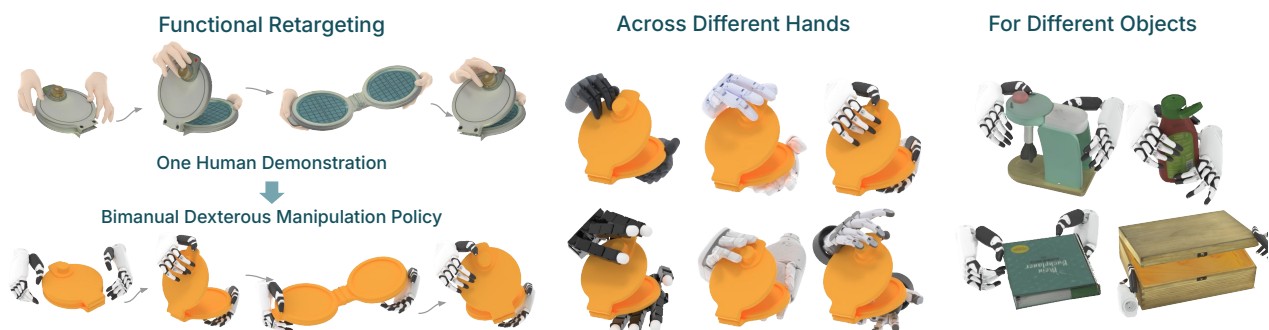

*Figure 1.* **Functional Retargeting.** We study the problem of functional retargeting, where the goal is to retarget human hand demonstrations into functional dexterous robot policies that manipulate an object to follow the demonstrated trajectory. Our proposed algorithm, DexMachina, achieves functional retargeting from one human demonstration to a variety of existing dexterous hand embodiments over a range of articulated objects.

hands and 5 articulated objects (Fan et al., 2023), and provides a unified testbed where new hands and tasks can be easily added and quickly evaluated. On this benchmark, we empirically show that DexMachina significantly outperforms baseline methods, and applies successfully to a wide variety of hands, articulated objects, and long-horizon demonstrations.

With an effective algorithm and evaluation benchmark for functional retargeting, it is now possible to make functional comparisons across different hardware: informed by the policy learning performance, we obtain a meaningful measure for both the hands' functionality and readiness to learn from human guidance. This comparison is generalizable and accessible: our algorithm requires no hand-specific adaptations, and our task environments are fast to run and easy to customize. With the recent surge in the development of robotic hand hardware, we hope this functional comparison will be helpful for making informed decisions for both acquiring and designing new hands.

**Our contributions are summarized as follows:**

• We study **Functional Retargeting**, where we learn feasible dexterous manipulation policies from human hand-object demonstrations. We propose **DexMachina**, a novel algorithm for functional retargeting based on a curriculum over virtual object controllers and motion and contact guidance.

• We introduce the DexMachina benchmark with 6 curated dexterous hand assets and 5 articulated objects, for evaluating both different functional retargeting algorithms and robotic hand designs.

• We demonstrate DexMachina achieves state-of-the-art learning performance across a variety of robotic hands and tasks. Our simulation environments and learning algorithms will be open-sourced to facilitate future research.

## 2 Related Work

**Reinforcement Learning for Dexterous Manipulation.**
Reinforcement Learning (RL) has been used for dexterous manipulation tasks such as in-hand object orientation (Andrychowicz et al., 2020; Handa et al., 2023; Qi et al., 2023; Yin et al., 2023; Chen et al., 2023) and single-hand grasping (Lum et al., 2024; Caggiano et al., 2023; Luo et al., 2024; Mandikal & Grauman, 2021; Zhu et al., 2023; Yuan et al., 2024), but achieving more complex, longer-horizon manipulation remains challenging due to the burden of designing rewards to guide exploration for such tasks. Model-based methods have been applied to tasks such as ball dribbling (Shiokata et al., 2005) and Rubik's cube turning (Higo et al., 2018), but they require careful engineering for each object and task. In our work, we seek to study bimanual long-horizon tasks where it can be difficult to specify concrete goals or design RL rewards to guide exploration. This motivates our use of human demonstrations, which both act as a goal specification and provide guidance for how to solve the task. Simulation is a common tool to train dexterous hand policies (Rajeswaran et al., 2018) due to the high exploration cost of running RL on real hardware (Xu et al., 2022). Our simulation benchmark supports evaluation across several dexterous hands and diverse tasks defined by human demonstration data, in contrast to existing RL benchmarks (Bao et al., 2023; Company, 2025) for dexterous manipulation.

**Imitation Learning for Dexterous Manipulation.** Imitation learning (IL) is a compelling alternative to RL, since the use of demonstrations can mitigate or eliminate the burden of exploration, but it can require accurate on-robot action data that is challenging to capture for dexterous hands. Most existing approaches (Qin et al., 2023; Wang et al., 2024; Yang et al., 2024; Cheng et al., 2024; Shaw et al., 2024; Zhang et al., 2025) require setting up a teleoperation system customized for a particular robot hand embodiment. Human

hand data (such as videos) are another source of data. Prior work has used human hand data for learning rough grasp affordances (Mandikal & Grauman, 2020), improved retargeting (Park et al., 2025), or co-training with human hand data and teleoperation data (Shaw et al., 2022; Xu et al., 2023), but these approaches have been limited for short-horizon manipulation (mainly grasping). Instead, our work assumes access to a single tracked hand-object demonstration per task and uses the demonstration to guide RL training. Similar approaches have been used for humanoid locomotion (Peng et al., 2018), simple hand manipulation (Wang et al., 2023), and dexterous manipulation on short-horizon tasks (Chen et al., 2024; Li et al., 2025).

**Curriculum Learning.** It is common practice in optimization-based motion planning to warm-start an optimization from relaxed physical constraints, resulting in a better solution at convergence (Mordatch et al., 2012; Pang & Tedrake, 2021; Pang et al., 2023). This idea of learning using a curriculum, which moves from easier to more difficult problems, has been adopted by RL methods (Chiappa et al., 2024; Zhang et al., 2024). Some prior work uses this approach to relax physical constraints, such as allowing force before making contact (Mao et al., 2025) or relaxing gravity, friction, and constraint-solver parameters (Li et al., 2025). Our approach uses a curriculum over object dynamics, allowing the agent to gradually learn how to manipulate the object over time (Fig. 2).

## 3 Functional Retargeting Formulation

We define the *functional retargeting* problem as follows: given one object $\eta$, one human hand-object demonstration sequence $\mathcal{D}^\eta$, and a pair of dexterous robot hands $\zeta$, the goal is to learn a robot policy that can manipulate the object to track the demonstrated object states. More formally, one human demonstration $\mathcal{D}^\eta = \{G, H\}$ contains $T$ timesteps of densely tracked object states $G$ and hand poses $H$. We focus on articulated objects, hence the object states include both part pose and revolute joint angle values. At any timestep $t$, given an achieved object state $\hat{g}_t$ (position, rotation, and articulation) and the target object state from the demonstration $g_t = \{g_t^P, g_t^R, g_t^J\}$, we denote the distance function as $F$ (computes both rotation, position, and articulation joint error). The learned policy for $(\eta, \zeta)$ should minimize the accumulated tracking error across all timesteps: $\pi_\theta^{\eta,\zeta} = \mathrm{argmin}_\theta \sum_{t=1}^T (F(\hat{g}_t, g_t))$.

We use bimanual hand demonstrations represented by MANO (Romero et al., 2017), which tracks 6-DoF wrist poses and 21 fingertip keypoint positions, hence $H = \{H_{\text{left}}, H_{\text{right}}\}, H^{\text{left}} = \{H_{\text{left}}^{\text{wrist}} \in \mathbb{R}^{T \times 6}, H_{\text{left}}^{\text{fingertip}} \in \mathbb{R}^{T \times 21 \times 3}\}$, and 1-DoF articulated objects with one revolute joint, hence $G \in \mathbb{R}^{T \times 8}$.

## 4 Method

**Overview.** We propose DexMachina, a curriculum-based RL algorithm for functional retargeting. In §4.1, we begin by introducing the task reward, which encourages object tracking in but is insufficient for effective policy learning. In §4.2, we extract motion and contact information from demonstrations, which we use to define residual actions and auxiliary rewards. While these components improve learning, they still fall short in complex long-horizon tasks. This motivates our curriculum strategy, presented in §4.3, where we introduce an auto-curriculum based on virtual object controllers to achieve efficient functional retargeting across different dexterous hands.

### 4.1 RL Environment and Task Reward

We train reinforcement learning (RL) policy to achieve the functional retargeting task. An RL environment is constructed by pairing one demonstration $\mathcal{D}^\eta$ and one set of bimanual dexterous robot hands $\zeta$. At each timestep $t$, write $G_t = \{g_t^P, g_t^R, g_t^J\}$ for the recorded object position, rotation and joint angles at timestep $t$, and $\hat{G}_t = \{\hat{g}_t^P, \hat{g}_t^R, \hat{g}_t^J\}$ for the object's achieved states corresponding to each term. The task reward $r_{\text{task}}$ is the product of three terms measuring accuracy in each state component, encouraging balanced learning (Chen et al., 2024). Formally:

$$r_{\text{pos}} = \exp(-\beta_{\text{pos}} d_{\text{pos}}); \ d_{\text{pos}} = ||\hat{g}_t^T - g_t^T||_2$$

$$r_{\text{rot}} = \exp(-\beta_{\text{rot}} d_{\text{rot}}); \ d_{\text{rot}} = 2\cos^{-1}(|\langle \hat{g}_t^R g_t^R \rangle|)$$

$$r_{\text{ang}} = \exp(-\beta_{\text{ang}}); \ d_{\text{ang}} = ||\hat{g}_t^J - g_t^J||_2$$

$$r_{\text{task}} = r_{\text{pos}} * r_{\text{rot}} * r_{\text{ang}}$$

where $\beta_{\text{pos}}$, $\beta_{\text{rot}}$, and $\beta_{\text{ang}}$ are scalar weights that control the desirable error scale for each component.

### 4.2 Action Formulation and Auxiliary Rewards

Although task reward specifies desired object states, it does not provide useful information for *how* to achieve them. To address this, we (1) propose a hybrid action formulation, which constrains the wrist action space to align more with the human demonstrators; and (2) define auxiliary rewards, which guide the policy to follow the human's hand-object interaction strategy. As a preliminary, we first apply pre-processing on the demonstration data $\mathcal{D}^\eta$ to extract relevant motion and contact information.

**Data Pre-Processing.** Given $\mathcal{D}^\eta$ with $T$ timesteps, $N$ object parts, and a dexterous hand $\zeta$ with $J$ actuated joints and $K$ collision links, we first run a kinematics-only retargeting algorithm (Qin et al., 2023) that matches dexterous hand poses with human hand motion. Then we obtain:

1. **Collision-aware kinematic retargeted joints** $\mathcal{Q} \in$

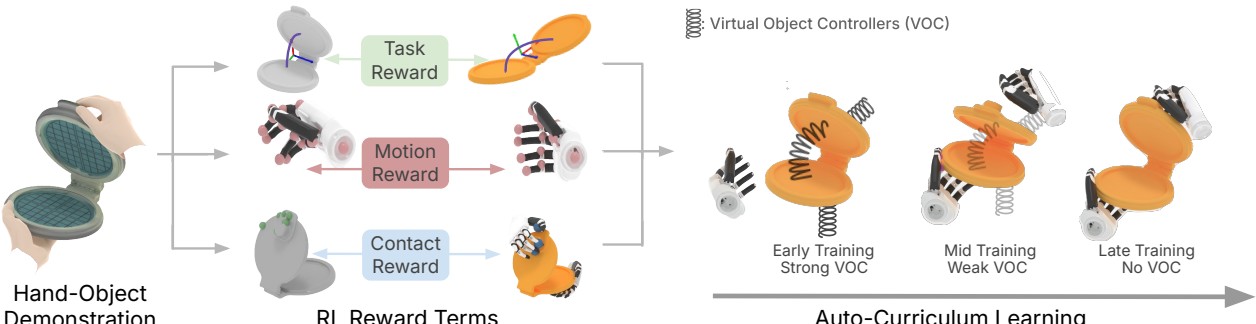

*Figure 2.* **DexMachina Overview.** DexMachina is a curriculum-based RL algorithm for functional retargeting. We process densely-tracked human hand demonstration to extract reference robot joints and keypoints (pink spheres) and approximated contact positions on object mesh vertices (green spheres), which we use to define auxiliary rewards in addition to the task reward. We then introduce an auto-curriculum using virtual object controllers, which initially moves the object on its own to follow the demonstration, and are then decayed over the course of RL training as the policy learns to take over manipulation.

$\mathbb{R}^{T \times J}$ and **reference keypoints** $\mathcal{X} \in \mathbb{R}^{T \times K \times 3}$ by replaying the retargeting results in simulation and recording (1) the achieved joint values and (2) 3D keypoint positions of the dexterous hand links. To eliminate object penetrations, we replay the retargeted joint values as soft control targets in simulation while keeping the object fixed — See Appendix B.2 for more details.

2. **Approximated hand-object contact**. Although kinematic retargeting produces human-like dexterous hand poses, the motions often fail to manipulate the object. Hence we extract contact information as additional guidance for object interaction. We use a distance-based approximation to obtain exactly when and where a specific dexterous hand link should be in contact with a specific object part (detailed in Appendix B.4). The results are approximated contact positions $C \in \mathbb{R}^{(T \times N \times K \times 3)}$ and a mask $M \in \mathbb{R}^{(T \times N \times K)}$ that indicates whether a pair of object part and hand link has valid contacts.

**Hybrid Action Outputs.** Given the retargeted joint results $\mathcal{Q}$, we use the joint values for 6-DoF wrist joints as base actions, which are added to the policy's output residual actions. The remaining finger joints use absolute actions that are normalized by their joint limits (see Appendix B.3 for full details). This formulation effectively constrains the policy's action space, and we empirically find it to significantly improve the learning efficiency.

**Motion Imitation Reward.** To encourage human-like hand motions, we take the motion reference keypoints $\mathcal{K}$ and retargeted joint values $\mathcal{Q}$, and define (1) motion imitation reward $r_{\text{imi}}$ based on keypoint matching, (2) behavior-cloning reward $r_{\text{bc}}$ based on joint angle distances to the reference. Formally:

$$r_{\text{imi}} = \frac{1}{K} \sum_{i=1}^{K} \exp(-\beta_{\text{imi}} ||\hat{x}_i - x_i||_2)$$

$$r_{\text{bc}} = \frac{1}{J} \sum_{i=1}^{J} \exp(-\beta_{\text{bc}} ||\hat{q}_i - q_i||_2)$$

where each $(\hat{x}_i, x_i)$ denotes the achieved and reference positions for the $i$th keypoint and $(\hat{q}_i, q_i)$ denotes the achieved and retargeted values for the $i$th joint.

**Contact Reward.** We read contact positions between each hand link and each object part, and compute contact reward by matching the policy contacts with the corresponding demonstration contacts. For each side of the hand, we denote the policy's and demonstration's contact positions and validity masks as $C, \hat{C} \in \mathbb{R}^{N \times K \times 3}$, $M, \hat{M} \in \mathbb{R}^{N \times K \times 1}$, respectively. We compute $L_2$ contact distance masked by validity masks and use it to define contact reward $r_{\text{con}}$:

$$r_{\text{con}} = \frac{1}{2NK}(\sum_{i=1}^{N} \sum_{j=1}^{K} \exp(-\beta_{\text{con}} D_{\text{left}}^{(i,j)}) + \exp(-\beta_{\text{con}} D_{\text{right}}^{(i,j)})) \quad (1)$$

where $D_{\text{left}}, D_{\text{right}} \in \mathbb{R}^{N \times K}$ and we set

$$D^{(i,j)} = \begin{cases} d_{\max}, & \text{if } M_{\text{demo}}^{(i,j)} \neq M_{\text{policy}}^{(i,j)} \\ 0, & \text{if } M_{\text{demo}}^{(i,j)} = M_{\text{policy}}^{(i,j)} = 0 \\ ||C^{(i,j)} - \hat{C}^{(i,j)}||_2 & \text{else} \end{cases} \quad (2)$$

The final RL reward is a weighted sum of the above terms: $r_t = \lambda_{\text{task}} r_{\text{task}} + \lambda_{\text{imi}} r_{\text{imi}} + \lambda_{\text{bc}} r_{\text{bc}} + \lambda_{\text{con}} r_{\text{con}}$. See Appendix B.4 for precise weights and additional reward details.

### 4.3 Auto-Curriculum with Virtual Object Controllers

**Motivation.** The above reward terms and action constraints are sometimes sufficient short and simple tasks, but struggle

**Algorithm 1** DexMachina Curriculum

---

**Require:** Reward thresholds $\sigma_{\text{task}}, \sigma_{\text{imi}}, \sigma_{\text{bc}}, \sigma_{\text{con}}$; Reward deques $D_{\text{task}}, D_{\text{imi}}, D_{\text{bc}}, D_{\text{con}}$
**Require:** Initial gains $k_p, k_v$, decay ratios $\phi_p, \phi_v$; Max episode length $L_{\max}$
  **for** each PPO iteration **do**
    **for** each environment where episode is done **do**
      Get: achieved episode length $L$, cumulative rewards $R_{\text{task}}, R_{\text{imi}}, R_{\text{bc}}, R_{\text{con}}$
      **for** each term $z \in \{\text{task}, \text{imi}, \text{bc}, \text{con}\}$ **do**
        Compute normalized reward: $\bar{r}_z = \frac{R_z}{L_{\max}}$
        Append $\bar{r}_z$ to deque $D_z$
      **end for**
    **end for**
    **for** each reward type $z \in \{\text{task}, \text{imi}, \text{bc}, \text{con}\}$ **do**
      Compute mean: $\mu_z = \text{mean}(D_z)$
    **end for**
    **if** $k_p = 0$ **then**
      **continue** // no need to decay
    **end if**
    **if** $\mu_z > \sigma_z \; \forall z \in \{\text{task}, \text{imi}, \text{bc}, \text{con}\}$ **then**
      // Learning is stable, applying gain decay
      $k_p \leftarrow k_p \cdot \phi_p$
      **if** $k_p \leq 0.01$ **then**
        $k_p \leftarrow 0; \quad k_v \leftarrow 0$
      **end if**
      $k_v \leftarrow k_v \cdot \phi_v$
    **end if**
  **end for**

---

on long-horizon clips with complex contacts. The policy often experiences catastrophic early-stage failures: e.g. after lifting a box with both hands, it might fail to anticipate that one hand will need to reposition mid-air to open the lid while the other hand adjusts for single-handed grasping. The policy would attempt different actions, most of which would drop the box and terminate the episode.

This motivates us to propose our curriculum approach, to let the policy explore different strategies in a less fragile setting. Our core idea is using *virtual object controllers*: they drive the object to follow the targets on its own, such that the policy can learn through the entire sequence and be discouraged from myopic strategies.

**Virtual Object Controllers.** We treat the demonstration states $G$ as control goals and apply virtual spring-damper constraints that move the object along its target trajectory. Initially, the virtual controllers handle most of the object movement; over time, the controller's influence is gradually reduced, requiring the policy to assume greater control to complete the task. They controllers are implemented using privileged information in simulation. Each object is equipped with six virtual 1-DoF joints for its base pose and

a 1-DoF joint for articulation, and all joints are actuated by PD controllers (Franklin et al., 2002). At every timestep, these controllers apply virtual forces based on the error between the current object state and control targets from the demonstration. The control strength is parametrized by gain parameters $(k_p, k_v)$, which are decayed over time to enable a structured hand-off to the learned policy.

**Curriculum scheduling.** Algorithm 4.2 describes our proposed curriculum. At the beginning of curriculum training, we set high virtual controller gains with critical damping; then we exponentially decay the gains based on the policy's learning progress, which is tracked with a history of past rewards. As a result, the policy initially will consistently achieve high task reward; because it receives a weighted sum of task and auxiliary rewards, the policy learns actions that improve motion and contact rewards while avoiding disrupting the object trajectory. Later, as the object controllers weaken, the policy gradually learns to adjust its motions to maintain high task reward. Because the auxiliary rewards use a much smaller weight, the policy can deviate from the reference hand motions learned at the earlier stages in order to prioritize optimizing for high task rewards.

## 5 Experiments

**Experiment Setup.** We use hand-object data from ARC-TIC (Fan et al., 2023) (see §B), which includes 5 articulated objects (Xu et al., 2025) and 7 demonstrations consisting of diverse motion sequences (picking up and reorienting objects, opening/closing lids, etc.) We evaluate our algorithm on both short- (used in prior work (Chen et al., 2024)) and long-horizon demonstrations. We curate assets for 6 open-source dexterous robot hand models, with varying sizes and kinematic designs. We use Genesis (Authors, 2024) for physics simulation, and PPO (Schulman et al., 2017; Makoviichuk & Makoviychuk, 2021) as the base RL algorithm. The policies share the same structure for state-based input observation spaces for all hands and tasks, and control both hands at once. See Appendix for details on RL training (§C.1) and evaluation setup §C.4.

**Baseline Methods.** Due to various differences in physics simulation and training configurations, we re-implement baseline methods in our training framework and make several adaptations to ensure a fair comparison — see § C.3 for implementation details. We compare against the following methods:

1. **Kinematics Only.** Directly use kinematic retargeting (Qin et al., 2023) results as control targets.

2. **ObjDex (Chen et al., 2024).** learns a high-level wrist planner for wrist base actions, and a low-level policy with task reward and the same hybrid actions as

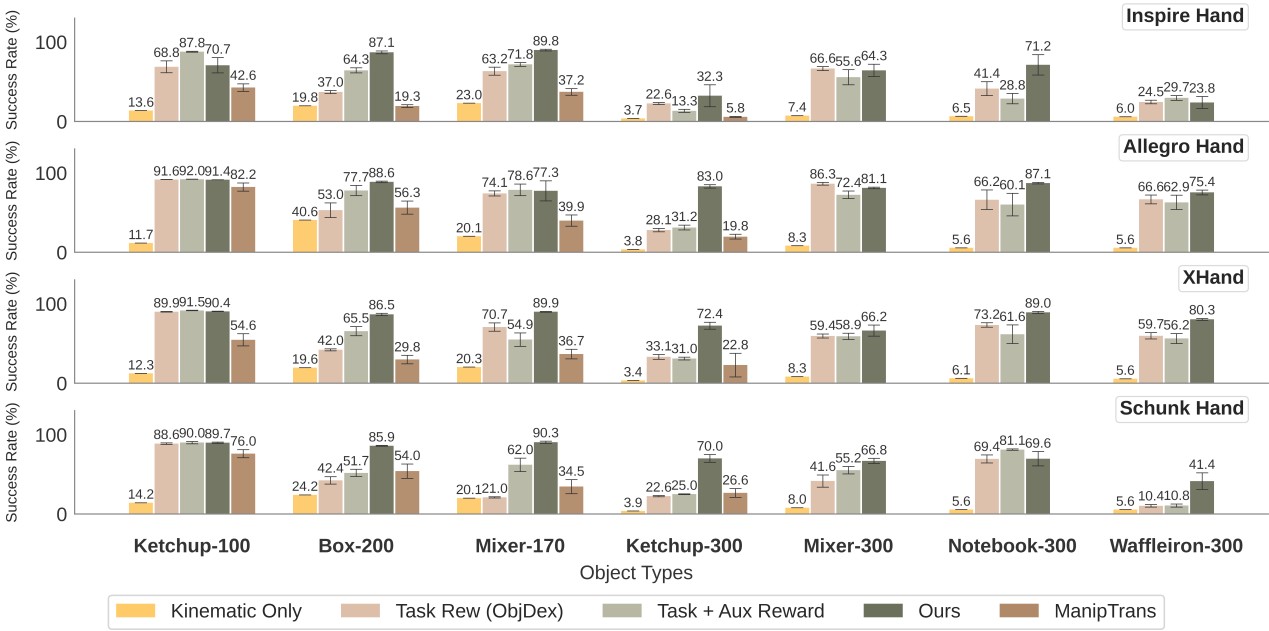

*Figure 3.* **DexMachina Core Results.** We evaluate DexMachina on four representative dexterous hands paired with seven demonstrations with diverse objects and motion sequences. We compare between direct replay of kinematic retargeting results ("Kinematic Only"), training with only a task reward ("Task Rew (ObjDex)", i.e., our re-implementation of ObjDex (Chen et al., 2024)), training with both task and auxiliary rewards ("Task + Aux Reward"), and with our proposed auxiliary rewards and curriculum ("Ours"). With rare exceptions, DexMachina demonstrates clear improvements over baseline methods, especially on long-horizon tasks with more complex motions.

ours. We validate our re-implementation by showing improved performance on the same demonstrations used in the original results.

3. **Task + Auxiliary Rewards without curriculum.** To evaluate the effect of our proposed curriculum, we run RL training with only our proposed motion imitation and contact rewards. For a fair comparison, all training hyper-parameters are identical with our curriculum setting.

4. **ManipTrans (Li et al., 2025).** A concurrent work that fine-tunes a motion imitation model with RL using contact force rewards and a curriculum on error thresholds and physics parameters. Since the original method is evaluated on rigid objects in a different physics simulator, we re-implement their proposed curriculum while using our hybrid actions and auxiliary reward terms.

**Overview of Experiments.** We present empirical results that (1) evaluate the effectiveness of DexMachina against baselines and no-curriculum settings (§ 5.1); (2) ablate key components of our method (§ 5.2); (3) demonstrate DexMachina's applicability across various dexterous hand embodiments and utility as an evaluation framework for comparing different hand designs (§5.3).

### 5.1 DexMachina Main Results

We evaluate DexMachina and baseline methods on four representative dexterous hands (Inspire (Technology, 2025), Allegro (WONIK-Robotics, 2025), Xhand (ROBOTERA, 2025), and Schunk (KG, 2025)) and seven demonstration clips (see 8 for visualization). Averaged success rates for each task are reported in Figure 3. The key takeaways are the following:

**DexMachina consistently improves performance on all hands and tasks.** We highlight the rightmost four columns in Fig. 3, which correspond to long-horizon demonstrations with complex motion sequences[2]. Task reward alone falls short on these clips; incorporating auxiliary rewards ('Task + Aux Rewards') improves performance on some tasks, but the gains are inconsistent. We also show that kinematic retargeting results alone cannot complete the task ('Kinematics Only') — our videos qualitatively show that they visually align well with human hands, but the actions cannot achieve more than slightly lifting up each object. n contrast, DexMachina significantly outperforms no-curriculum setting despite using the same rewards.

---

[2]For instance, 'Waffleiron-300' requires the policy to pick up the object, open and close the lid, flip it back and forth, then open and close the lid again, all mid-air (see Appendix §8 for a visualization)

**DexMachina successfully handles complex long-horizon dexterous tasks.** On short-horizon tasks (i.e. around 100 timesteps), task reward and hybrid actions can achieve reasonable performance: our re-implementation of Ob-jDex (Chen et al., 2024) ('Task Rew (ObjDex)' in Fig. 3) performs better than their original reporting on the same demonstrations (left three columns in Fig. 3, detailed in §C.3). We observe that the benefit of our algorithm is more prominent as we move to longer-horizon tasks. The main experiments use human demonstrations with up to 300 timesteps (10 seconds real time), and we provide additional limited-scale results that qualitatively show DexMachinacan train policies for even longer demonstrations of up to 600 timesteps. We remark that this ability to learn much longer tasks is due to our virtual object controller design, which enables the policy to explore the entire demonstration clip without getting stuck in early failures, which the baseline methods suffer from.

**DexMachina lets the policy learn task strategies that adapt to hardware constraints.** The auxiliary rewards do not always align with the best task strategy, but instead act as soft guidance to serve the curriculum, giving the policy flexibility to explore. Qualitatively, we observe that the policies may deviate from the motion and contact guidance and learn different strategies: as shown in Fig. 4: on Notebook-300, the XHand policy follows the human demonstrator to use the left hand to hold up the object and the right hand to close the cover; however, for the smaller, less-actuated Inspire Hand, the policy learns to use both hands to stabilize the object and close the cover. On Mixer-300, the Allegro Hand fingers are long enough to close the lid easily, but the Schunk Hand policy shows more wrist movements to achieve the same effect.

### 5.2 DexMachina Ablations

**Action Ablations.** We compare our hybrid action formulation with: (1) absolute actions on all joints and (2) less-constrained residual actions on wrist joints, in which the wrist joint limits are set to cover the maximum motion range in the entire demonstration clip. We train in the no-curriculum setting, use a subset of tasks and hands and average over three seeds for each method. Results are shown in Fig. 5. While all methods benefit from using auxiliary rewards, using more restrictive bounds on wrist motion results in the best overall performance.

**Curriculum Ablations.** In Fig. 3, we compare Dex-Machina with ManipTrans (Li et al., 2025), which uses a curriculum over error thresholds for motion and object poses plus gravity and friction parameters. We observe that it achieves no clear improvements over the no-curriculum setting, and training is less stable: given the same budget of RL iterations, the ManipTrans policy initially achieves

high task reward, but performance drops as the curriculum progresses and cannot recover. This indicates that merely decaying physics parameters is not sufficient for long-horizon tasks with articulated objects, which needs a stronger guidance to completely solve the task until the policy gradually takes over.

**Reward Weights Ablations.** Although we use a weighted sum of several different reward terms, we provide additional experiments to show that policy performance is relatively insensitive to the weight hyper-parameters as long as they are within a reasonable range. since the maximum achievable reward for both the task reward and each auxiliary reward term is always 1.0, we consistently use 1.0 for the task reward and ensure that the weighted sum of auxiliary rewards does not exceed 1.0. More experimental details are provided in Appendix D.1.

**VOC Curriculum Hyper-parameter Sweeps.** We test the sensitivity of our RL training to different hyperparameters in our VOC curriculum, and show that the curriculum decay is robust to small variations in decay reward thresholds, dequeue lengths for reward tracking, and the decay ratio for VOC controller gains. Full details and results are on our submission website: `dexmachina-submission.github.io`

### 5.3 Hand Embodiment Analysis

After validating that DexMachina achieves functional retargeting across various tasks and hands, we now use our algorithm and benchmark for a functional comparison between different dexterous hands. We focus on the four long-horizon tasks from §5.1, and evaluate DexMachina on two additional hands, Ability (PSYONIC, 2025) and DexRobot Hand (see Fig. 6). We discuss the following key findings:

**Larger, fully-actuated hands achieve both higher final performance and better learning efficiency.** The Allegro Hand, despite being less anthropomorphic in appearance, is surprisingly capable due to its long finger length providing stability for in-hand / in-air manipulations.

**Similarity in size is less important than degrees of freedom.** For instance, the Inspire, Ability, and Schunk Hand all have similar sizes, but Schunk has actuated fingertips and a foldable palm, and achieves on average better performance than Inspire and Ability.

**Although less-actuated hands are more similar to human hands in appearance, learned strategies are less human-like than the bigger but more capable hands.** Because all hands use the same set of human hand motion references (both as base wrist actions and motion rewards), the extent to which a policy deviates from human guidance is determined by their size and kinematic constraints. As a result, hands like Inspire and Ability often need different strategies to

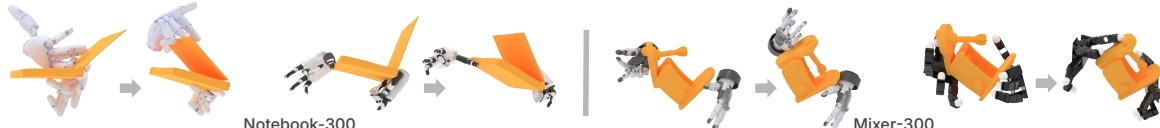

*Figure 4.* **DexMachina Learns Hand-Specific Task Strategies.** DexMachina enables the policy to learn task strategies that adapt to their hardware constraints. We show snapshots of trained policy rollouts for different hands on the same task: left side shows XHand and Inspire Hand for Notebook-300 task; right side shows Schunk Hand and Allegro Hand for Mixer-300 task.

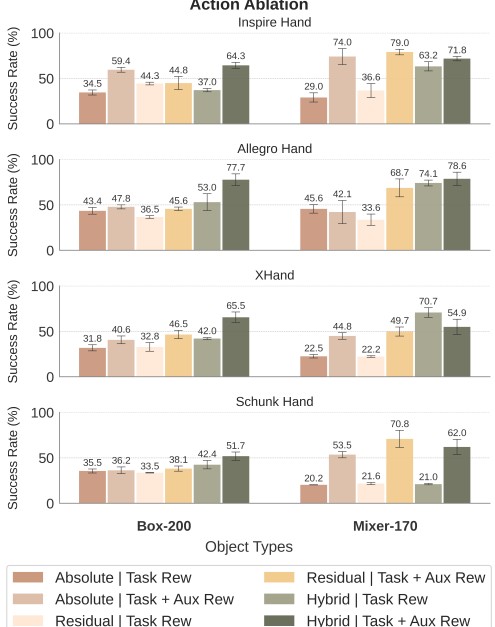

*Figure 5.* **Hand Action Ablation.** We ablate on action output formulations on a subset of dexterous hands and objects and trained *without* curriculum. Hybrid actions with more restrictive bounds (light and dark green bars) shows better learning performance than absolute actions and full residual actions with less wrist constraints, both in training with task rewards or with both task plus auxiliary rewards settings

complete the task.

Naturally, our conclusions are limited by the objects and tasks that we test on: for instance, the larger hands will not perform well for smaller objects (e.g., tweezers). However, our evaluation framework can be easily extended to add new dexterous hands and test tasks or objects.

## 6  Limitations

DexMachina has a few key limitations that we leave for future work. First, our policy uses privileged information from the physics simulation. To enable better transfer to the real world, our algorithm can either be extended to vision-based RL policy training, or supplemented with a distillation step that trains visuomotor policies using demonstration data generated from our policies – both routes have been

## Evaluation of DexMachina Across All Hands

| | Mixer | Notebook | Ketchup | Waffleiron |
|---|---|---|---|---|
| Ability | 28.1 ±7.4 | 51.6 ±10.1 | 9.0 ±0.6 | 9.1 ±0.7 |
| Inspire | 64.3 ±7.8 | 71.2 ±12.9 | 32.3 ±13.8 | 23.8 ±7.5 |
| Schunk | 66.8 ±3.3 | 69.6 ±9.0 | 70.0 ±4.8 | 41.4 ±10.5 |
| Dexrobot | 73.9 ±2.0 | 73.5 ±3.6 | 36.4 ±12.0 | 33.8 ±8.4 |
| XHand | 66.2 ±7.0 | 89.0 ±1.2 | 72.4 ±4.4 | 80.3 ±1.3 |
| Allegro | 81.1 ±1.0 | 87.1 ±1.0 | 83.0 ±2.2 | 75.4 ±3.0 |

*Figure 6.* Full evaluation of all six hands using DexMachina. We focus on four long-horizon tasks from §5.1. Empircal results suggest that: 1) Larger, fully-actuated hands achieve both higher final performance and better learning efficiency. 2) Similarity in size is less important than degrees of freedom. 3) Although less-actuated hands are more similar to human hands in appearance, learned strategies are less human-like than the bigger but more capable hands.

validated in prior work (Lum et al., 2024; Chen et al., 2024).

Second, we use open-source assets for the dexterous hands and estimate physical properties (such as mass, inertia, and collision shapes), the simulated hands might fail to capture some of the dynamics of the real hardware. More careful tuning with real reference hardware or more accurate models provided by manufacturers will be needed to address this.

Lastly, our dexterous hand policies are trained with 'floating' 6 DoF wrists. This allows us to empirically compare policy learning for different dexterous hands without being biased by arm kinematics. In the supplementary materials, we qualitatively show a learned policy's wrist poses can be achieved by robot arms via Inverse Kinematics.

## 7  Conclusion

We present DexMachina, a curriculum-based RL algorithm for functional retargeting, using virtual object controllers that aid policy exploration under motion and contact guidance. In our simulation benchmark with a diverse set of tasks and dexterous hands, we show DexMachina significantly outperforms baseline methods and enables functional comparison across different dexterous hand designs. We

hope our algorithm and benchmark environments will provide a useful platform for identifying desirable dexterous hand capabilities and lower the barrier for contributing to future research.

## 8   Impact Statement

The goal of this paper is to advance the fields of machine learning and robot learning, specifically in dexterous manipulation learned from human demonstrations. The primary impact of this work is enabling more systematic evaluation and comparison of dexterous hand capabilities, which may benefit applications in manufacturing, assistive robotics, and household automation.

As with many advances in robotic manipulation, there are potential risks associated with unsafe behavior when transferring learned policies from simulation to real-world systems. These risks can be mitigated through careful validation, safety constraints, and responsible deployment practices. We do not foresee direct applications of this work to surveillance, weapons, or privacy-invasive technologies. Overall, we believe the anticipated benefits to research and engineering outweigh the potential risks.

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

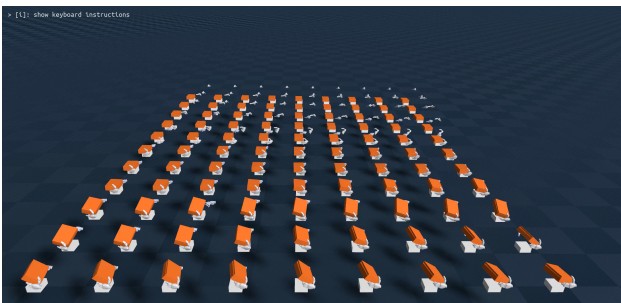

*Figure 7.* We perform an improved retargeting scheme over pure kinematic retargeting

# A  Appendix

**For additional qualitative result videos, please see our submission website:**
**project-dexmachina.github.io**

# B  Demonstration Data Processing Details

## B.1  ARCTIC Demonstration Selection and Curation

We use a subset of hand-object interaction clips from the ARCTIC dataset (Fan et al., 2023), which contains articulated object scans and interaction sequences with tracked MANO (Romero et al., 2017) hand poses and object states. Each selected clip is defined by an object (e.g. 'box'), a subject tag (e.g. 's01-u01') identifying the human demonstrator, and a (start, end) tuple to trim the sequence to a fixed length, hence the number of used frames $T$ is defined as $T = (\text{end} - \text{start})$

**Dexterous Hand Asset Processing.**  All of the dexterous hands in our experiments are curated from open-source URDF models and manually edited to add 6-DoF wrist joints that achieve a 'floating hand' style wrist actuation. Some of the dexterous hand models require additional processing for stable simulation, such as manually changing mass or inertia values, running convex-decomposition to improve collision mesh quality, and adding dummy links to fingertips to record and track keypoint positions. For each dexterous hand, we manually specify which finger links should match with which MANO (Romero et al., 2017) hand joints(e.g. thumb to human thumb), which is required by the kinematic retargeting (Cheng et al., 2024) algorithm. The kinematic retargeting results are also used for controller gain tuning, which ensures the dexterous hand controllers are stable and fast enough to match the desired human hand movement and speed within reasonable error.

## B.2  Object-aware Retargeting Post-processing

Because we use densely-tracked human hand and object interaction as demonstration, a purely kinematic retargeting

algorithm (Cheng et al., 2024) on fingertip positions results in frequent penetration with the object, which leads to damaging base-actions during policy learning, and infeasible keypoint positions which we use for imitation reward computation. To address this, we run the simulation for each pair of dexterous hands, and for each demonstrated timestep, we fixate the object to its target state (both root pose and object joint angle), and set retargeted joint values as control targets. This process lets the simulation to resolve collision and

Then we record the achieved joint values and keypoints to use for policy learning. In implementation, this process can be easily parallelized in simulation, which we illustrate in Figure 7.

## B.3  Hybrid Action Outputs.

Formally, we use the following notations:

- clip($x, a, b$): elementwise clamp of input value $x$ between $a$ and $b$

- $a_t \in \mathbb{R}^J$: the policy's joint action output at time $t$ clipped to $[-1, 1]$, i.e. $a_t = \text{clip}(\pi_\theta(o_t), -1, 1)$

- $q_t^{(i)}$: the target position for the $i$-th joint at time $t$

- $\mathcal{I}_f \subset \{1, \ldots, J\}$: indices corresponding to the finger DOFs

- $\mathcal{I}_w^{\mathrm{T}} \subset \{1, \ldots, J\}$: indices of the three wrist translation DOFs, $|\mathcal{I}_w^{\mathrm{T}}| = 3$

- $\mathcal{I}_w^{\mathrm{R}} \subset \{1, \ldots, J\}$: indices of the three wrist rotation DOFs, $|\mathcal{I}_w^{\mathrm{R}}| = 3$

- $\mathbf{q}_t \in \mathbb{R}^J$: the **retargeted** joint values at time $t$

- $s_{\mathrm{T}}, s_{\mathrm{R}}$: scaling factors for translation and rotation actions respectively

- $\ell, u \in \mathbb{R}^J$: vectors of lower and upper joint limits

- $\hat{q}_t \in \mathbb{R}^J$: the joint target values sent to the policy's controller

Then, the joint target computation is defined as:

$$a_t^{\text{wrist-T}} = a_t[\mathcal{I}_w^{\mathrm{T}}] \in \mathbb{R}^3, \quad q_t^{\text{wrist-T}} = \mathbf{q}_t[\mathcal{I}_w^{\mathrm{T}}] + s_{\mathrm{T}} \cdot a_t^{\text{wrist-T}}$$
$$a_t^{\text{wrist-R}} = a_t[\mathcal{I}_w^{\mathrm{R}}] \in \mathbb{R}^3, \quad q_t^{\text{wrist-R}} = \mathbf{q}_t[\mathcal{I}_w^{\mathrm{R}}] + s_{\mathrm{R}} \cdot a_t^{\text{wrist-R}}$$
$$a_t^{\text{fingers}} = a_t[\mathcal{I}_f], \quad q_t^{\text{fingers}} = \ell_{\mathcal{I}_f} + \frac{u[\mathcal{I}_f] - \ell[\mathcal{I}_f]}{2} \cdot (a_t^{\text{fingers}} + 1)$$
$$\hat{q}_t = \text{concat}(q_t^{\text{wrist-T}}, \ q_t^{\text{wrist-R}}, \ q_t^{\text{fingers}})$$

### B.4 Contact Approximation

Let: $V_o = \{v_i^o\}_{i=1}^{N_o}$ be the vertices of one object part mesh, $V_h = \{v_j^h\}_{j=1}^{N_h}$ be the vertices of one MANO hand mesh, $\gamma$ be the contact distance threshold, $N_c$ be the maximum number of raw contact approximations (we use $\gamma = 0.01, N_c = 50$), and $K$ be the number of collision links on a dexterous robot hand.

First, we do contact approximation by finding object mesh vertices that, their $L_2$ distance to the nearest neighbor on the MANO mesh is within $\gamma$: for each $v_i^o$, we get $v_j^* = \arg\min_j \|v_i^o - v_j^h\|_2$, and mark $v_i^o$ as an approximate contact point if $\|v_i^o - v_{j*}^h\|_2 < \gamma$. When there's more than $N_c$ vertices within this threshold, we use farthest subsampling to get the final set of $N_c$ contacts, denoted as $C = \{v_k^c\}_{k=1}^{N_c} \subset V_o$.

Next, we "retarget" the raw approximate contacts to the dexterous robot hand: let $L = \{\ell_m\}_{m=1}^K$ be the center positions of the dexterous hand links, for each contact point $v_k^c \in C$, assign it to the nearest link: $m^* = \arg\min_m \|v_k^c - \ell_m\|_2$ For each link $\ell_m$, compute the average position of the assigned contacts: $\bar{v}_m = \frac{1}{|C_m|} \sum_{v_k^c \in C_m} v_k^c$ where $C_m \subset C$ is the subset of contacts assigned to link $\ell_m$. If $|C_m| = 0$, then $\bar{v}_m$ is marked as invalid.

The final outputs are:

- A contact tensor $\mathcal{C} \in \mathbb{R}^{T \times N \times K \times 3}$

- A validity mask $\mathcal{M} \in \{0, 1\}^{T \times N \times K}$

where $T$ is the number of time-steps in the demonstration clip, $N$ is the number of object parts ($N = 2$ for all our articulated object assets). The exact same procedure is repeated for each dexterous hand, hence each bi-manual RL task environment has two copies of contact information with the same shapes.

## C Experiment Details

### C.1 RL Training and Evaluation Details

We use Genesis for physics simulation (Authors, 2024) and PPO as the base RL algorithm implemented by the rl-games (Makoviichuk & Makoviychuk, 2021) package. In the reported results for both ours and baseline methods, we use $12,000$ parallel environments for RL training on all the dexterous hands, except for Dex Hand which uses $10,000$ environments due to memory constraints. Each training run occupies either one single NVIDIA L40s or H100 GPU, and we run 5 random seeds for each demonstration and each pair of dexterous hands for all compared methods, except for action ablation experiments in §5.2 which use 3 random seeds.

### C.2 RL Policy Observation and Action Space

We use state-based input for policy observation space: this include object states, joint targets, finger-to-object distances, and normalized hand-object contact forces.

### C.3 Details on Baseline Reimplementation

Our most relevant baseline methods (Chen et al., 2024; Li et al., 2025) are built with Isaac-Gym (Makoviychuk et al., 2021) with various simulation-specific implementation details. To ensure a fair comparison, we have dedicated effort to ensure a faithful reimplementation using our training framework and RL environments. Some of our modifications can result in better performance for the baselines than their original reports: for example, Genesis (Authors, 2024) uses a more stable simulation contact modeling and is more memory efficient, which enables training up to $12,000$ parallel environments with much higher learning efficiency than the Isaac Gym environments used by baselines (i.e. $2048$ for ObjDex (Chen et al., 2024) and $4096$ environments for ManipTrans (Li et al., 2025)). We describe further details our reimplementation for each baseline below:

**ObjDex (Chen et al., 2024) Reimplementation Details.** To ensure a fair comparison, we have contacted the original authors to obtain their setup details that were not available publicly, including: 1. A good estimate for the exact frame start- and end- parameters for the ARCTIC clips used for training; 2. A frame interpolation multiplier that effectively extends the episode length for RL training to be longer than the original demonstration (e.g. an ARCTIC clip with $T$ timesteps requires training RL on $4T$ or $7T$ episode steps). We reuse their clip range but choose not to use the interpolation after empirically finding it to increase training time without improving task performance.

Moreover, the original ObjDex (Chen et al., 2024) method uses a two-level framework, where a high-level wrist planner is first learned across all ARCTIC demonstration clips, and a low-level RL policy outputs wrist residual actions. We instead directly use the kinematic retargeting results for the wrist base actions. The high-level wrist planner design assumes access to a bigger dataset and makes the low-level RL policy sensitive to the learned planner outputs — **we hypothesize this is the main reason for why our reimplementation can achieve better performance than the original results** (e.g. On Ketchup-100, our re-implementation achieves $> 90\%$ success rate for all hands, whereas the original paper reports $41.2\%$; on Mixer-170, ours achieves $> 70\%$ success rate for three out of four hands, whereas the paper reports $57.6\%$).

**ManipTrans (Li et al., 2025) Reimplementation Details.** Because the original method did not directly evaluate on

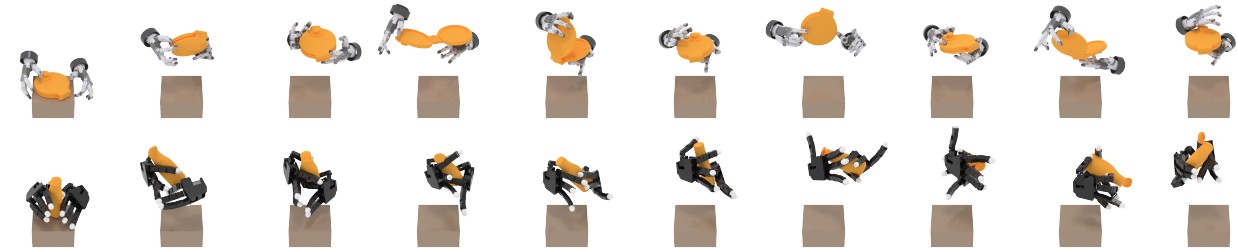

*Figure 8.* Visualization of the long-horizon tasks achieved by our trained RL policy. The brown box is used as platform surface, which follows the original ARCTIC data collection setup where objects are placed on a square cardboard box on a table surface (Fan et al., 2023).

ARCTIC demonstrations (albeit the paper's appendix Section A.1 (Li et al., 2025) reported *qualitative* results for some ARCTIC objects), we reimplement their proposed curriculum while keeping everything else aligned with our best-performing setup (this includes hybrid action formulation, training with both task and auxiliary rewards and the same RL hyper-parameters, etc.). We follow the original method (Li et al., 2025) to decay four parameters during training, namely the thresholds for object pose errors and hand keypoint error, the z-axis gravity value, and the friction parameter, which we write as $\epsilon_{\text{object}}^{P}, \epsilon_{\text{object}}^{R}, \epsilon_{\text{finger}}, g_{\text{gravity}}, \mu$, respectively. We modified Genesis (Authors, 2024)'s rigid solver to support modifying the gravity vector during RL training. ManipTrans (Li et al., 2025) does not disclose the exact decaying schedule for these parameters or the range for gravity and friction parameters, hence we choose the same exponential scheduler to stay consistent with our virtual object controller curriculum, and choose a range of $g_{\text{gravity}} \in [0, -9.81], \mu \in [4.0, 1.0]$. More specifically, given a max iteration $I$, desired range of parameters, and a decay interval $v$, the parameter is decayed every $v$ iterations; after the parameters reach their final values, training proceeds for another fixed number of iterations (this is also aligned with our method). The exponential schedule depends on the given max iterations $\mathcal{I}$, for each parameter $\omega \in \{\epsilon_{\text{object}}^{R}, \epsilon_{\text{finger}}, g_{\text{gravity}}, \mu\}$: its value at a given training iteration can be written as $\omega_{\text{current}} = \omega_{\text{init}} \cdot \left(\frac{\omega_{\text{final}}}{\omega_{\text{init}}}\right)^{t/I}$. Note that we use a pseudo value $\bar{g}_{\text{gravity}} \in [9.81, 0]$ because the decay computation assumes positive bounds, and the actual applied gravity is $g_{\text{gravity}} = 9.81 - \bar{g}_{\text{gravity}}$.

### C.4 Policy Evaluation Setup

**Evaluation Across Random Seeds.** For each method and task, we run 5 random seeds; each seed run saves a best policy checkpoint based on cumulative task reward, and each checkpoint is evaluated for 20 episodes. For each evaluation episode, we record the achieved object states (both pose and revolute joint angle) and compare against the demonstration trajectory.

**Performance Metrics.** Our functional retargeting task requires a manipulation policy to achieve articulated object tracking, which involves balancing both pose and joint angle errors over long time sequences. For performance reporting, prior work has explored per-step success rate (Chen et al., 2024) or tracking error (Li et al., 2025), but both have clear shortcomings: success rate reporting is based on how many timesteps out of the entire demonstration a policy can move the object to track within given error thresholds, hence the results are highly sensitive to the threshold values (this case requires 3 different thresholds for position, rotation, and joint angle errors), which also depend on the object size and geometries. Reporting tracking errors can be accurate, but it shows three different errors for every task, hence making it difficult to derive high-level comparisons and takeaways from experiment results. To address these limitations, we propose to follow prior work in object pose tracking (Wen et al., 2024; Xiang et al., 2018; Hinterstoisser et al., 2013) and use a similar ADD-AUC[3] metric with the key difference that we compute ADD for each object part separately (to accommodate articulated objects) and average ADD results before computing AUC. We found this to be a less-sensitive metric that still reports one success rate value for each method while reflective of the qualitative results from policy roll-outs.

## D  Additional Experiment Results

We visualize keyframes of our long-horizon manipulation tasks in Fig. 8. Please see supplementary videos for additional qualitative results for policy roll-outs. The figure in the next page shows results for our action ablation experiments described in §5.2.

### D.1  Reward Weights Ablation

In the experiments below, we vary the weights for motion imitation, behavior cloning and contact rewards, while keep-

---

[3]ADD stands for Average Distance, AUC stands for Area Under Curve, we don't use ADD-S because we have the exact matching targets from the demonstration

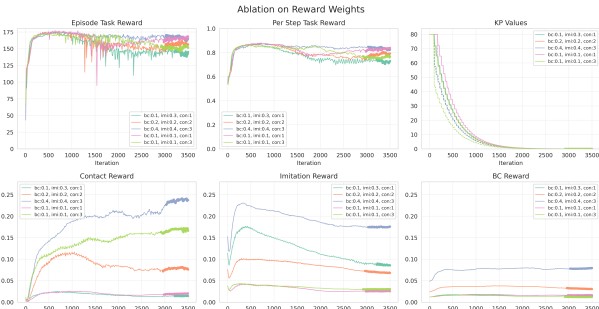

*Figure 9.* **Reward Weights Ablation.**

ing other settings exactly the same. Each curve uses a different combination of reward weights, and we plot the task reward (which directly reflects object tracking performances) and auxiliary rewards. The sub-plot 'Episode Task Reward' plots the cumulative task reward achieved in each episode, which directly reflects the policy's object tracking performance. Although we could not finish an exhaustive pass on all the possible weight combinations due to the limited bandwidth during this rebuttal period, we remark the overall trend that the task performance is relatively robust to different auxiliary reward weights. The slight fluctuations in the Episode Task Reward curves will likely be smoothed out when averaged across multiple random seeds. We provide further detail in supplementary website.

