# OpenReview forum: "DexMachina: Functional Retargeting for Bimanual Dexterous Manipulation"
_ICML.cc/2026/Conference — ICML 2026 regular_

### Official Review · Reviewer_bAzx · 2026-03-10

**Soundness:** 3
**Presentation:** 3
**Significance:** 3
**Originality:** 3
**Overall Recommendation:** 4
**Confidence:** 4

**Summary:**

This paper introduces DexMachina, a curriculum-based reinforcement learning algorithm designed for functional retargeting, the process of learning dexterous robot manipulation policies from human hand–object demonstrations. The method addresses challenges in long-horizon, bimanual manipulation tasks involving articulated objects, which are difficult due to large action spaces and the mismatch between human and robot embodiments. DexMachina incorporates virtual object controllers that initially guide objects along demonstrated trajectories; as their influence gradually decreases, the policy learns to take over through motion and contact cues. Alongside the algorithm, the authors present the DexMachina benchmark, a simulation platform featuring six robotic hands and five articulated objects to standardize the evaluation of dexterous manipulation and hardware design. Experimental results show that DexMachina outperforms baseline methods, enabling functional comparisons across robotic hardware and providing a flexible, accessible platform to advance research in dexterous manipulation learning.

**Compliance With Llm Reviewing Policy:**

Affirmed.

**Key Questions For Authors:**

- How do the authors plan to adapt DexMachina for real-world deployment given its reliance on privileged simulation information?
- How sensitive is the algorithm’s performance to the curriculum schedule and hyperparameter settings, such as the decay rate of virtual controller strength?
- How robust is DexMachina to variations in the quality or noise of human demonstration data used for training?

**Limitations:**

yes

**Strengths And Weaknesses:**

### Strength:

1. This paper tackles an important and challenging problem: functional retargeting. To the best of the reviewer’s knowledge, hand–object interactions involving articulated objects and dexterous robotic hands are particularly difficult due to the rich contact dynamics and complex physical interactions.
2. The paper, including the presentation of the main algorithm, is clear and easy to follow. Readers can quickly grasp the key contributions and understand the design and motivation of the proposed method.
3. The experimental results, supported by an accompanying video, convincingly demonstrate the promising empirical performance of the proposed approach across different embodiments. The work has the potential to influence and inspire future research in this area.
4. The proposed benchmark is a valuable contribution that can facilitate further development in the field. It provides a standardized and accessible platform for evaluating dexterous manipulation algorithms and hardware designs.

### Weakness
1. A key limitation of this work is the lack of real-world experiments to demonstrate the transferability of the proposed method beyond simulation. As noted in the paper’s limitations section, the algorithm relies on privileged information available only in physics simulations, which significantly constrains its applicability in real-world scenarios where such data are inaccessible. Consequently, the learned policy may struggle or fail when deployed on physical systems. This limitation is especially critical given the paper’s goal of informing real-world hardware design, as the absence of physical validation limits the practical impact and generalizability of the proposed approach.

2. The algorithm involves multiple stages of curriculum learning, and the transitions between these stages can be highly sensitive to the choice of hyperparameters. These sensitivities may significantly affect the final performance, suggesting that training with DexMachina may not be as straightforward as the paper implies. Careful tuning and calibration of the curriculum schedule are likely required for stable learning and successful policy transfer across different tasks and environments.

3. While the proposed method demonstrates promising performance, its novelty appears somewhat limited. The algorithm primarily builds upon standard reinforcement learning frameworks combined with curriculum learning, which have been explored in prior studies. Although the integration is well-executed and effective for the functional retargeting problem, the conceptual advancement beyond existing approaches is relatively incremental rather than fundamentally new.

---

> ### Author Rebuttal · Authors · 2026-03-30
>
> Thank you reviewer bAzx for the detailed review and constructive feedback. We hope the following response will help clarify our method details and address the raised concerns.
>
> ### - Real world applicability and connection to hardware design.
>
> Our method supports common sim-to-real transfer approaches, and prior works using similar training setting has [2,3] demonstrated successful real world results, through either pose-estimation to deploy state-based policies [2], carefully designed motion replay [3], or distilling a robust policy from teacher policies [1]. Since our method achieves better simulation results than these baselines, we believe real world deployment is promising following similar methods.
>
> In our submission website, we also showed 1) the feasibility of connecting our hand policies to robot arms via IK, and 2) adding randomization to our RL training can further improve a policy’s robustness to varying physics dynamics, both are key factors in real world deployment. Concretely, the next step to real world transfer is 1) setting up camera-based object pose tracking system for providing policy input; 2) adding wrist IK which obtains joint actions for bimanual robot arms.
>
> ### - Hardware evaluation.
>
> We note our method has as different focus for evaluation of hardware design: instead of evaluating physical hardware capabilities such as payload or durability, our RL training pipeline evaluates the _learning efficiency_ of a given hardware design, i.e. how fast and how well the policy can learn from human data. We believe our method is informative and helps such evaluations, and simulation provides a more accessible platform for cross-embodiment comparisons.
>
> ### - Sensitivity to hyper-parameters.
>
>
> We would like to clarify that our experiments used the same hyper-parameters across different tasks and objects and did not require task-specific tuning. We provided additional experiment results to test the sensitivity for some key hyper-parameters – due to the page limit here, we refer the reviewers to our experiments section in paper and submission website for more details and visualizations: https://dexmachina-submission.github.io/#curriculum-hyperparams
>
> - *Decay reward thresholds.* We keep a separate deque and threshold for each reward term (i.e. task reward, contact reward, imitation reward), and only decay the curriculum’s VOC controller gains when the achieved rewards are above the given threshold.
>
> - *Curriculum rewards dequeue lengths.* We compare different deque lengths (30, 60, 90) under the same task and robot hand setting. Intuitively, a longer deque means the moving reward average gets updated more slowly. Hence we observe overall similar task reward outcomes, but slightly longer training time when the deque length is longer.
>
> - *Decay Ratios for VOC controller gains.* In our VOC curriculum, the controller gain values (i.e. kp and kv) are drawn from a uniform distribution with decaying upper and lower bounds. The bounds are exponentially decayed at every iteration where the rewards are above their respective thresholds. We showed that the task performance (i.e. object tracking accuracy) is relatively insensitive to different bounds, but a ‘tighter’ bound results in lower performance, which suggests learning is more robust if the policy is exposed to a wide range of different VOC dynamics.
>
> ### - Algorithmic novelty.
>
> Our proposed virtual object controller curriculum is novel by design and enables long-horizon dexterous policy learning. We hope it still provides useful insights and demonstrates new capability from simulation-based RL learning.
>
> [1]] Ankile, L., Simeonov, A., Shenfeld, I., Torne, M. and Agrawal, P., 2024. From Imitation to Refinement--Residual RL for Precise Assembly. arXiv preprint arXiv:2407.16677.
>
> [2] Y. Chen, C. Wang, Y. Yang, and C. K. Liu. Object-centric dexterous manipulation from human motion data. ArXiv, abs/2411.04005, 2024.
>
> [3] K. Li, P. Li, T. Liu, Y. Li, and S. Huang. Maniptrans: Efficient dexterous bimanual manipulation transfer via residual learning. arXiv:2503.21860, 2025.

---

> > ### Author Rebuttal · Reviewer_bAzx · 2026-04-02
> >
> > I have gone over the rebuttal. While my concern over the novelty and real-world impact has not been fully solved, this work still has some meaningful impact, and I will keep my rating.

---

### Official Review · Reviewer_TuTX · 2026-03-12

**Soundness:** 2
**Presentation:** 3
**Significance:** 3
**Originality:** 3
**Overall Recommendation:** 3
**Confidence:** 4

**Summary:**

This paper presents DexMachina, a policy learning framework for dexterous manipulation that aims to transfer human demonstrations across dexterous hands with different embodiments. The method is trained with a curriculum-based reinforcement learning strategy and introduces virtual object controllers to simplify the early stage of learning, before gradually shifting toward the final manipulation policy. The experiments cover six dexterous hand configurations and several manipulation tasks, and the reported results show improvements over baseline methods.

**Compliance With Llm Reviewing Policy:**

Affirmed.

**Key Questions For Authors:**

In addition to the virtual object controllers, DexMachina also includes specially designed auxiliary rewards and a hybrid action formulation. However, the paper does not appear to provide a sufficiently detailed ablation study to clarify the contribution of each component. It would be helpful to isolate the effect of these design choices more carefully.

The current experimental setup differs substantially from real dexterous manipulation scenarios. In particular, the dexterous hands are evaluated without being mounted on robot arms, which makes the setting less realistic. The paper would be stronger with additional transfer-oriented experiments, such as sim-to-sim validation under more realistic dynamics, or preferably real-robot experiments.

**Limitations:**

Yes

**Strengths And Weaknesses:**

Strengths:

The paper is built on a reasonable and important intuition. For dexterous manipulation, the central issue is not simply reproducing hand motion, but maintaining stable and accurate control of the manipulated object. Framing the problem in this way is appropriate and gives the method a clear motivation.

Another positive aspect is the breadth of the experimental setting. The method is evaluated on multiple dexterous hand embodiments and several different tasks, which is useful for examining how well the approach adapts across hand designs. The results suggest that DexMachina performs better than baselines such as Kinematics Only and ObjDex, and the experiments also provide some insight into how different hand morphologies affect learning and task execution.

Weaknesses:

My main concern is that the algorithmic novelty is somewhat limited. The key component of the method is the use of virtual object controllers, but this idea is not entirely new and has appeared previously in the literature, especially in curriculum-based learning for control. In this paper, the component is adapted to dexterous manipulation, which is useful, but the conceptual advance over prior controller-assisted learning methods appears modest.

A second concern is that the experimental validation is still limited. Although the simulation results are encouraging, the current setup remains relatively controlled and does not fully reflect the complexity of real dexterous manipulation. In particular, the experiments do not include any real-robot validation. Since the paper positions the method as a practical framework for dexterous hand learning, the lack of hardware results reduces the strength of the empirical claims. At a minimum, stronger transfer-oriented validation would make the paper more convincing.

---

> ### Author Rebuttal · Authors · 2026-03-30
>
> Thank you to reviewer 1r1F for the detailed review and constructive feedback. We hope the following response will help clarify our method details and address the raised concerns.
>
> ### - Algorithmic novelty.
>
> To our best knowledge, our proposed virtual object controller curriculum is a novel design in the dexterous manipulation literature. We hope it still provides useful insights and demonstrates new capability from simulation-based RL learning applied to long-horizon dexterous policy learning. Regarding prior work on “controller-assisted learning methods”, we would greatly appreciate any pointers to relevant literature to cite and improve our related work section.
>
>
> ### - Real world deployment
>
> Our framework can be combined with commonly-used sim2real approaches as seen in prior works, such as using pose-estimation to deploy state-based policies [3], carefully designed motion replay [4], or distilling a robust policy from teacher policies [2]. Since our method achieves better simulation results than these baselines, we believe real world deployment is promising following similar methods. In our submission website, we also showed 1) the feasibility of connecting our hand policies to robot arms via IK, and 2) adding randomization to our RL training can further improve a policy’s robustness to varying physics dynamics, both are key factors in real world deployment.
>
> ### - Question on auxiliary reward design and action formulation.
>
> Please refer to results for these ablations in Experiment section 5.2, Figure 5, and our submission website https://dexmachina-submission.github.io/#reward-weights. Overall, we remark that 1) auxiliary rewards, i.e. motion imitation, keypoint tracking, and contact matching, are not conceptually too different from standard practice in prior works like [3,4], but differ in implementation details and we carefully process the kinematic retargeting data to improve the reference quality. The role of these rewards in VOC curriculum is more for guidance, and the policy should always prioritize the task reward. 2) residual action improves learning efficiency a lot, intuitively because the policy needs to explore a much smaller action space, and this finding is consistent with prior works [3,4].
>
>
> ### - Question on arm mounting feasibility and generalization across different physics dynamics.
>
> In our submission website we showed that 1) inverse kinematics (IK) can be used to find arm joint solutions that support our learned polices’ wrist actions: https://dexmachina-submission.github.io/#arm-ik. 2) domain randomization can be added to our RL training https://dexmachina-submission.github.io/#randomize and nudge the policy to using more stable contact between hand and task objects.
>
>
>
> [1] Ankile, L., Simeonov, A., Shenfeld, I., Torne, M. and Agrawal, P., 2024. From Imitation to Refinement--Residual RL for Precise Assembly. arXiv preprint arXiv:2407.16677.
>
> [2] Y. Chen, C. Wang, Y. Yang, and C. K. Liu. Object-centric dexterous manipulation from human motion data. ArXiv, abs/2411.04005, 2024.
>
> [3] K. Li, P. Li, T. Liu, Y. Li, and S. Huang. Maniptrans: Efficient dexterous bimanual manipulation transfer via residual learning. arXiv:2503.21860, 2025.

---

> > ### Author Rebuttal · Reviewer_TuTX · 2026-04-04
> >
> > I still believe that experimental verification in real-world environments is crucial, but the author's rebuttal fails to demonstrate this point.

---

### Official Review · Reviewer_1r1F · 2026-03-12

**Soundness:** 4
**Presentation:** 4
**Significance:** 3
**Originality:** 4
**Overall Recommendation:** 5
**Confidence:** 4

**Summary:**

In this work, the authors proposed DexMachina, a curriculum-based reinforcement learning method for functional retargeting for bimanual dexterous manipulation, which utilizes virtual object controllers and motion and contact guidance. In addition, they introduced a benchmark with six dexterous hand assets and five objects for the evaluation of algorithms and robotic hand designs. The experimental results demonstrate that DexMachina significantly outperforms baselines with an ablation study on a subset of dexterous hands and objects and trained without curriculums.

**Compliance With Llm Reviewing Policy:**

Affirmed.

**Final Justification:**

The paper addressed a meaningful problem with solid work. My recommendation will be 5: Accept.

**Key Questions For Authors:**

1. would the authors kindly provide more insights about the sim-to-real performance gap?

**Limitations:**

Yes

**Strengths And Weaknesses:**

---------------------------------Strength----------------------------------
1. Overall, the paper is well-written, and the presentation is very good.
2. The intuition is well-explained and easy to follow.
3. The experiments are comprehensive and look promising.

---------------------------------Weakness---------------------------------------
1. All the validations were conducted in a simulation, which makes sense for the curriculum-based setup, but the learning outcome would still benefit from a real robot study.
2. All the objects used in the study were hard objects, and no deformable objects were involved.
3. While the manipulation tasks with multiple steps are in general considered as long-horizon tasks, it is a bit weak for the paper to highlight the long-horizon tasks as a major contribution. I would suggest weakening that claim a bit or adding a task with multi-stages.

---

> ### Author Rebuttal · Authors · 2026-03-30
>
> Thank you to reviewer 1r1F for the detailed review and constructive feedback. We hope the following response will help clarify our method details and address the raised concerns.
>
>
> ### - Path to real world deployment
> Our framework can be combined with commonly-used sim2real approaches as seen in prior works [1,2,3], such as using pose-estimation to deploy state-based policies, carefully designed motion replay, or distilling a robust policy from teacher policies. Since our method achieves better simulation results than these baselines, we believe real world deployment is promising following similar methods. In our submission website, we also showed 1) the feasibility of connecting our hand policies to robot arms via IK, and 2) adding randomization to our RL training can further improve a policy’s robustness to varying physics dynamics, both are key factors in real world deployment.
>
>
> ### - Handling deformable objects.
>
> We would need two main steps to apply our method to deformable objects. First is building a real-to-sim pipeline that can import real world deformable object manipulation demonstrations into a physics simulation that supports non-rigid physics (which the Genesis simulation engine we use does support). Second is updating our pose-based task reward function (since 6DoF pose tracking does not directly apply) – this can be replaced with 3D keypoint matching, i.e. reward the policy for matching 3D keypoints to the demonstrated  motion trajectory.
>
>
> ### - Claim on long-horizon tasks.
>
> Thank you for the feedback, we will update the paper manuscript to emphasize that we focus more on long sequences of continuous motions and better distinguish from multi-stage, more semantically rich manipulation tasks.
>
>
> [1] Ankile, L., Simeonov, A., Shenfeld, I., Torne, M. and Agrawal, P., 2024. From Imitation to Refinement--Residual RL for Precise Assembly. arXiv preprint arXiv:2407.16677.
>
> [2] Y. Chen, C. Wang, Y. Yang, and C. K. Liu. Object-centric dexterous manipulation from human motion data. ArXiv, abs/2411.04005, 2024.
>
> [3] K. Li, P. Li, T. Liu, Y. Li, and S. Huang. Maniptrans: Efficient dexterous bimanual manipulation transfer via residual learning. arXiv:2503.21860, 2025.

---

> > ### Author Rebuttal · Reviewer_1r1F · 2026-04-03
> >
> > I appreciate the authors' in-depth response.
> >
> > Most of my concerns are well explained. I believe the work is solid and will have a meaningful contribution to the field.

---

### Official Review · Reviewer_WoV2 · 2026-03-12

**Soundness:** 3
**Presentation:** 3
**Significance:** 3
**Originality:** 3
**Overall Recommendation:** 4
**Confidence:** 4

**Summary:**

This work propose training a robot manipulation model that retargets a given human hand-object interaction demonstration into feasible robot hand motions to achieve robust hand-object interaction. The RL training is based on a task reward and auxiliary rewards (motion and contact guidance). To address the instability of direct RL training on complex tasks, they introduce an auto-curriculum that uses virtual object controllers to physically guide the object during the early stages of training. Experiments show that this approach yields better overall performance compared to baselines, especially in bimanual, and long-horizon tasks.

**Compliance With Llm Reviewing Policy:**

Affirmed.

**Key Questions For Authors:**

1. In the comparison to ManipTrans, whether the general policy of ManipTrans is pretrained across all tasks and only the residual action policy is tailored for each demonstration. Since the general policy of ManipTrans is expected to be pretrained across all tasks in their paper and then frozen, (1.1) how can it be shown that the comparison is fair?  (1.2) Or is only the curriculum part implemented rather than the full ManipTrans general–residual architecture? (1.3 ) How much impach does this have on the results in comparison?


2.  (2.1) Why not include more than one demonstration for a single task, or train a general policy across different tasks instead of training them separately for each demonstration? (2.2) How long does a single RL run take?  (2.3) Isn’t this computationally too expensive for a large demonstration dataset to be retargeted?

**Limitations:**

Yes

**Strengths And Weaknesses:**

Strength:

* Novel Curriculum Design: The introduction of an auto-curriculum using the norvel Virtual Object Controllers to physically guide the object during early-stage training is a creative and effective solution for overcoming early exploration failures.

* The experimental setup is solid and extensive. The authors thoroughly validate the algorithm effectiveness across diverse tasks of six different dexterous robot hands and a wide variety of complex, bimanual manipulation tasks.

Weakness:

* Lack of Task Generalization: The method relies on training a highly specialized policy from scratch for just one single hand-object demonstration. The paper does not demonstrate whether a single trained policy can generalize across multiple different human demonstrations of the same task, limiting its robustness to variations in human movement.
Similarly lack of generalization ability using a same policy for different tasks, (Unlike ManipTrans, which has a generalist mode works for all tasks and need only 15 min to retrain the residual action policy for each demonstration to save training effort.). If given a large-scale dataset, the retargeting can takes too long using this method that require retrain from scratch for each demonstrations.

* No real-world application is demonstrated. Additionally, not all observation states used in this work are accessible in real-world settings. For example, finger–object distances and object states that need to be estimated are not directly available; in practice, only images are typically available. The approach could be made more practical for real-world scenarios.

---

> ### Author Rebuttal · Authors · 2026-03-30
>
> Thank you to reviewer WoV2 for the detailed review and constructive feedback. We hope the following response will help clarify our method details and address the raised concerns.
>
>
> ### - Task Generalization
> Our empirical results demonstrated that learning long-horizon single tasks is challenging, which we aim to address in this work. Further scaling to more multi-task data is indeed an exciting future direction. The ARCTIC dataset we used contains high quality data but for only 11 objects with highly variant geometries and object-specific hand motions, which makes it unlikely for multi-task training to enable generalization to unseen tasks at the current scale. If given more data, a more practical solution to scaling up would be training many single-task policies using DexMachina the distill into a more general multi-task policy.
>
>
> ### - Real world applicability
> For sim-to-real transfer, there are proven alternatives to RGB images that have smaller sim2real gaps, such as object poses and depth images, which our policy can use for observation input. Our trained RL policies can be combined with common sim2real techniques as seen in prior works, such as using pose-estimation to deploy state-based policies [3], carefully designed motion replay [4], or distilling a robust policy from teacher policies [2]. Since our method achieves better simulation results than these baselines, we believe real world deployment is promising following similar methods.
>
> To obtain the object-to-fingertip observation in a real world setup, one could use segmentation models combined with either depth cameras or estimated depth to obtain object pointcloud, which is similar to our implementation in simulation, i.e. sample the object vertices positions and compute distance to fingertips, and proven effective in prior work [1].
>
> ### - Comparison to ManipTrans.
> In ManipTrans, the general hand motion imitation policy is first trained on multi-task kinematic demonstrations, then for each single-task RL policy, the general policy provides the base action at every timestep, so the RL policy learns a residual action on top of that base action. Their general policy has seen more tasks, but it was trained with only hand tracking and smoothness rewards (no object tracking) – hence the base actions it produces should be no better than retargeting results from our collision-aware kinematic retargeting step (although, for unseen demonstrations, if this imitation policy generalizes, it could potentially be more efficient than re-running kinematic retargeting).
>
> In our setup, the RL policy also learns residual actions, but the base action directly comes from our collision-aware kinematic retargeting pipeline. We compare to the RL+curriculum part of ManipTrans and feed it the same high-quality base actions that our VOC curriculum also uses. Therefore we compare only the curriculum part (1.2) and believe the comparison is fair (1.1). Because the general imitation policy from ManipTrans can ‘at best’ produce the same base actions as kinematic retargeting, we think this setup benefits the ManipTrans RL policy instead of harming it in our comparison.
>
> ### - Question regarding single-task formulation.
>
> 2.1 This is mainly a result of implementation design choice: because we found the RL policy learning is a lot more efficient with residual actions, for each RL training run we need to store (T,) timesteps demonstration data to compute rewards and provide base actions, this include a dense list of target object poses, robot hand keypoints, reference actions, and contact positions, which is easier to implement in an RL run as single task. But it should be straightforward to extend it to multi-task RL (with more memory and longer training time for each RL run), or distill multiple single-task policies into one.
>
> 2.2. Because we use a VOC curriculum that tracks policy success and terminates early, the training runs take between 4 to 12 hours depending on task difficulty. Although slower than kinematic retargeting, our framework produces an autonomous close-loop policy that outputs physically feasible actions and handles randomized physics dynamics in sim. This setup is commonly used in prior works [3], and our VOC curriculum significantly simplifies training as compared to methods like DexTrack [5], where each policy for each demonstration requires multiple iterations of retraining that each strictly depends on the prior training run.
>
>
>
> [1] Lin, Toru, et al. “Sim-to-Real Reinforcement Learning for Vision-Based Dexterous Manipulation on Humanoids”
>
> [2] Ankile, L., et al. "From Imitation to Refinement--Residual RL for Precise Assembly."
>
> [3] Y. Chen, et al. "Object-centric dexterous manipulation from human motion data."
>
> [4] K. Li, et al. "Maniptrans: Efficient dexterous bimanual manipulation transfer via residual learning."
>
> [5] Liu X, et al. "DexTrack: Towards Generalizable Neural Tracking Control for Dexterous Manipulation from Human References"

---

> > ### Author Rebuttal · Reviewer_WoV2 · 2026-04-03
> >
> > Thanks for the rebuttal, but the concerns remain unresolved:
> >
> > * Task Generalization: I am not referring to multiple tasks, but at least generalization across a single task with different demonstrations, since the trajectories of each demonstration for a single task have some variance. This aspect has not been sufficiently evaluated with numerical results.
> >
> > * Real-world applicability: Depth cameras can also suffer from occlusion, and object-to-fingertip observations can be noisy. Even small input variations may lead to task failure, since finger–object interactions require high precision. This raises concerns about robustness outside controlled simulation environments.
> >
> > * Comparison to ManipTrans: The authors use collision-aware kinematic retargeting instead of a pretrained general policy, and claim that the general policy of ManipTrans is no better than retargeting results from collision-aware kinematic retargeting. However, no numerical experiments are provided to support this claim, so the comparison with ManipTrans appears somewhat unfair.
> >
> > * Single-task formulation: Similar to the task generalization issue, even for single tasks, no experiments demonstrate performance across multiple trajectories of the same task. Additionally, 4–12 hours per trajectory is unrealistic for retargeting a large dataset.
> >
> > * Additional concern: In the provided videos, such as the Notebook and Mixer tasks, the robot hands sometimes use only one or two fingertips or edges to hold the object, rather than using the palm as in human demonstrations. While this may work in simulation, such grasps with only fingertips or edges are likely to be highly unstable in real-world settings, any slight disturbance (as in the real world) could cause complete failure. Therefore, the retargeted trajectories may be very unrobust to disturbances and may only work in perfect simulation environments while achieving high ADD-AUC scores.

---

> > > ### Author Response · Authors · 2026-04-08
> > >
> > > Thank you for the additional feedback.
> > > - If by 'task generalization' the reviewer meant generalizing across different hand motions on the same object + semantics (e.g. multiple human data clips could be picking up a box but with different pickup locations), we did not evaluate this in the paper, but we think it's straightforward to combine our single-clip policies with a multi-clip distillation setup described above and achieve this level of generalization.
> > > - Indeed the fingertip-to-object observation term might be noisy, hence it would make more sense to use such inputs for oracle RL policy training and distill into depth or segmented depth inputs as used in the referenced prior works.
> > > - Because the general policy from ManipTrans was trained with `hand tracking and smoothness rewards`, it is supervised to at-best predict these guided kinematic-only hand motions. Even if the general policy can perfectly predict such motions from training set, it could under-fit certain hand keypoints (due to robot-vs-human hand size differences) and cause object penetrations because it was trained on hand motion only. It is indeed difficult to numerically compare the quality of these base actions because they get combined with RL residual action learning efficiency, hence we think it's more fair to use the same base actions and only compare our curriculum RL algorithm with the curriculum proposed in ManipTrans.
> > > - Regarding the RL policy might learn unstable contact strategies that are not robust for sim-to-real transfer: it is a valid concern and we would add additional domain-randomization terms to further randomize the simulation physics to prevent such overfitting (although we do not have the bandwidth at this stage to re-run these experiments). Additionally we remark that the policy transferability is also impacted by hardware capabilities and should require less tuning as better hardware becomes available -- recent works [1] have demonstrated successful transfers of more dynamic and flexible hand motions with state-of-the-art hardware -- but this is a bit beyond our current scope.
> > >
> > > [1] Kedia, Kushal, et al. "SimToolReal: An Object-Centric Policy for Zero-Shot Dexterous Tool Manipulation." arXiv preprint arXiv:2602.16863 (2026).

---

### Decision · Program_Chairs · 2026-04-30

**Decision:**

Accept (regular)

**Comment:**

This paper proposes DexMachina, a functional retargeting framework for bimanual dexterous manipulation that combines virtual object controllers with curriculum learning. While the reviewers acknowledged the experimental breadth across six simulation tasks, multiple reviewers noted that the core method is relatively straightforward, combining reward shaping with auto-curriculum, and that all experiments are conducted entirely in simulation without real-world validation. These concerns remained partially unresolved after the rebuttal. We encourage the authors to strengthen the contribution with real-world experiments and resubmit to a future venue.